# Efficacy and Safety of Sesame Oil Cake Extract on Memory Function Improvement: A 12-Week, Randomized, Double-Blind, Placebo-Controlled Pilot Study

**DOI:** 10.3390/nu13082606

**Published:** 2021-07-28

**Authors:** Su-Jin Jung, Eun-Soo Jung, Ki-Chan Ha, Hyang-Im Baek, Yu-Kyung Park, Soog-Kyoung Han, Soo-Wan Chae, Seung-Ok Lee, Young-Chul Chung

**Affiliations:** 1Clinical Trial Center for Functional Foods, Jeonbuk National University Hospital, Jeonju 54907, Korea; sjjeong@jbctc.org (S.-J.J.); esjung@jbctc.org (E.-S.J.); soowan@jbnu.ac.kr (S.-W.C.); solee@jbnu.ac.kr (S.-O.L.); 2Biomedical Research Institute of Jeonbuk National University Hospital, Jeonju 54907, Korea; 3Healthcare Claims & Management Incorporation, Jeonju 54858, Korea; omphalos9121@hanmail.net (K.-C.H.); hyangim100@gmail.com (H.-I.B.); yukyungpark07@gmail.com (Y.-K.P.); 4Department of Food Science and Human Nutrition, Jeonbuk National University, 567 Baekje-daero, Jeonju 54896, Korea; skhan27@hanmail.net; 5Department of Internal Medicine, Division of Gastroenterology and Hepatology, Jeonbuk National University Medical School, Jeonju 54896, Korea; 6Department of Psychiatry, Jeonbuk National University Medical School, Jeonju 54896, Korea

**Keywords:** cognitive, sesame oil cake extract, memory function, β-amyloid, sesaminol

## Abstract

The goal of treatment for mild cognitive impairment (MCI) is to reduce the existing clinical symptoms, delay the progression of cognitive impairment and prevent the progression to Alzheimer’s disease (AD). At present, there is no effective drug therapy for AD treatment. However, early intake of dietary supplements may be effective in alleviating and delaying the MCI. This study aims to evaluate the effects of sesame oil cake extract (SOCE) supplementation on cognitive function in aged 60 years or older adults with memory impairment. A total of 70 subjects received either SOCE (*n* = 35) or placebo (*n* = 35) for 12 weeks based on random 1:1 assignment to these two groups. Cognitive function was evaluated by a computerized neurocognitive function test (CNT), and changes in the concentrations of plasma amyloid β (Aβ) proteins and urine 8-OHdG (8-hydroxy-2′-deoxyguanosine) were investigated before and after the experiment. Verbal learning test index items of the CNT improved markedly in the SOCE group compared to the placebo group (*p* < 0.05). Furthermore, plasma amyloid-β (1–40) and amyloid-β (1–42) levels in the SOCE group decreased significantly compared to that in the placebo group (*p* < 0.05). There was no statistically significant difference in urine 8-OHdG between the two groups (*p* > 0.05). Collectively, intake of SOCE for 12 weeks appears to have a beneficial effect on the verbal memory abilities and plasma β-amyloid levels of older adults with memory impairment.

## 1. Introduction

Mild cognitive impairment (MCI) is a relatively broad clinical state with mild memory impairment and a precursor to dementia of Alzheimer type (DAT) [1]. The prevalence of MCI was reported to be 10–20% in people over 65 years of age. Alzheimer’s disease (AD) risk increases with age and the risk is higher in men than in women [2]. Further, about 10% of the MCI cases progress to AD after one year, while 30–50% of cases move to AD after five years [3]. It is considered a much higher transition rate to AD than seen in individuals without MCI (1–2%) [4]. Neuropsychological symptoms and pathological changes accompany AD, and no effective treatment option is available. Thus, it is important to prevent and manage AD through early diagnosis at the MCI stage with no clinical symptoms [5,6]. The objectives of MCI treatment are to reduce the existing clinical symptoms, delay the progression of cognitive impairment, and prevent the AD. Unfortunately, there is no effective pharmacological therapy for MCI. However, dietary interventions may help to treat MCI. According to meta-analyses and epidemiological studies, the Mediterranean diet (Med diet), dietary approach to systolic hypertension (DASH), and Mediterranean-DASH Intervention for Neurodegenerative Delay (MIND) diet, which merges the Med diet and DASH diet, can reduce both the risk of AD and the progression of the MCI to AD [7,8,9,10]. Specifically, the Med diet is known to have a neuroprotective effect by reducing inflammation and oxidative stress. Therefore, specific dietary interventions may lower the risk of cognitive impairment. A recent study investigating natural and synthetic over-the-counter supplements in MCI and AD patients [11] reported the ineffectiveness of ω-3 fatty acids, soy, ginkgo biloba, folic acid, vitamins B, and antioxidative nutrient in protecting MCI and AD. However, phytochemicals have been reported to improve specific cognitive domains in AD patients [12]. Thus, more studies on the effects of natural compounds and functional foods on MCI and AD are necessary [13,14,15,16,17,18,19,20,21,22,23,24]. Sesame seeds (*Sesamun indicum* L.) are an important source of sesame oil in several countries. Sesame seeds contain phenolic lignans, which exhibits antioxidant actions in vitro and in vivo, regulate hyper-unsaturated fatty acid metabolism [25], detoxify the liver [14,15,26,27], and inhibit the absorption of cholesterol [16,28]. After roasting the sesame seeds at an appropriate temperature and extracting the oil by pressing, sesame oil cakes (SOCs) are obtained as a by-product. The major active ingredient in SOC is sesaminol glucoside (SG). Additionally, lignans including sesamin, sesamolin, and sesaminol from the decomposition of sesame fiber are present [29]. SOC is shown to protect against cognitive impairment in both in vitro and in animal models [30,31,32]. High SG-containing sesame oil cake extract (SOCE) effectively suppresses nerve cell apoptosis and inflammatory reactions caused by amyloid β (Aβ) induction and lipopolysaccharide exposure [30,31,32]. Hence, SOCE may be effective against memory damage and inhibiting memory loss caused by β-amyloid deposition [32]. However, there are no reports on the potential effects of SOCE on cognitive function in humans. This study determines the effects of SOCE on cognitive function in older adults (aged above 60 years) with memory impairment.

## 2. Materials and Methods

### 2.1. Participants

This study was conducted according to the guidelines of the International Conference on Harmonization Good Clinical Practice (ICH GCP) after reviewing the study and its approval by Jeonbuk National University Hospital Institutional Review Board (JUH IRB) and was implemented from 15 September 2017 to 31 January 2019 at the Clinical Trial Center for Functional Food (CTCF2) of JUH (IRB No. 2017-07-036). The entire human study was conducted in accordance with the provisions of the Helsinki Declaration, the standards for clinical trial management (IGCP) and the protocol was registered at www.clinicaltrials.gov (NCT03826121). Potential subjects completed signed informed consent and underwent a screening test. A total of 70 subjects were selected.

#### 2.1.1. Criteria for the Selection of Subjects Were as Follows:

Over 60 years of age at the time of the screening test.Subject capable of deciphering Korean.Subjects memory index scores that fell greater than 1 standard deviations (SDs) from the normal mean value for each test item in the neuropsychological part of (word list memory, word list recall, and word list recognition test) the Korean version of the Consortium to Establish a Registry for Alzheimer’s Disease Assessment Packet (CERAD-K) [23]. In addition, subjects who met at least one of the three test criteria for CERAD-K above.Written consent after being thoroughly educated about the study’s aims and goals.

#### 2.1.2. Subjects Who Met Any of the following Criteria Were Excluded from the Study: 

History of treatment for Axis I disorders within the last three years based on the Structured Clinical Interview for DSM-IV (SCID).Alcohol abuse or dependence within the last three months.Presence of any of the following diseases: epilepsy, mental retardation, cerebral nervous system disease, endocrine disease, blood malignancy, cardiovascular disease, and/or Crohn’s disease.Those who had any of the following abnormal laboratory results:AST, ALT > three times the upper limit of the normal range;Other abnormal laboratory results.Those who had taken any prescription medicine or herbal medicine within two weeks prior to the first day of intake or who had taken any general medicine (OTC) or vitamin supplements within one week.Those who had participated in other human studies within 2 months prior to the first day of intake.Those who had donated whole blood within a month prior to the first day of intake or who had donated a blood component within 2 weeks prior to the first day of intakeThose who were considered unfit for participation in this human study for any other reasons noted by the research manager.

### 2.2. Study Design and Randomization

This single-organization, random assignment, double-blind, and placebo-controlled human study was conducted for 12 weeks. The random assignment table was based on a sequence of random A and B numbers generated by the randomization module of SAS^®^ system version 9.2 (SAS Institute, Cary, NC, USA). SAS^®^ generated the code before the study was initiated. Subjects were registered after the first visit within four weeks prior to the screening test to review the suitability of the selection and exclusion criteria. Subjects were assigned in a 1:1 random manner to the SOCE group or placebo group to complete the baseline by the first visit date. Subjects were given the clinical study application products and asked to consume them before breakfast, lunch, and dinner three times a day for 12 weeks. In the sixth week, subjects were asked to visit the CTCF2, where vital signs, drug tolerance, medical conditions, and adverse reactions were noted, in addition to other tests being performed. Subjects were monitored as required either after the final intake of the clinical study application products or after the early completion. Three tests were performed to measure the drug’s effects on cognitive function. The first test was the screening. The second test (week 0) was conducted before the participants started taking the test products, and the third test (week 12) was conducted after completion of 12 weeks.

### 2.3. Preparation of Test Materials

The refined test products used in this study were provided by Keukdong H-Pharm Co., Ltd. (Yesan, Chungnam, Korea). Sesame oil cake (SOC), a byproduct of the sesame oil extraction process from sesame seeds. SOCE was extracted with water at 95 °C for 1 h to obtain a crude extract. The crude extract was further purified by Diaion HP 20 (Mitsubishi Chemicals, Co., Tokyo, Japan) column using ethanol. The extracts were concentrated under reduced pressure and freeze-dried. The yield rate of SOCE was 2%, and the concentration of sesaminol in the SOCE was 3.10 mg/g. The HPLC chromatogram of the SOCE is shown in Figure 1.

In our pilot clinical study, raw SOCE at 1.5 g/day is given to elderly subjects with mild cognitive impairment. The total score of the Korean Mini-Mental State Examination (K-MMSE) was significantly increased in the SOCE group than the control group. Moreover, plasma β-amyloid level was significantly decreased in the SOCE group than in the control group. Based on these pilot-scale observations and consumption patterns, 1.5 g of SOCE was set as the daily intake for this study. All subjects who participated in this study were randomly assigned between the SOCE and placebo groups. The SOCE group took a total of 3 g (Including 1.5 g/day as SOCE) per day, orally three times a day (before breakfast, lunch, and dinner). Similarly, the placebo group took 3 g (0 g/day as SOCE) orally by consuming one tablet before breakfast, lunch, and dinner every day. Each tablet given to the SOCE group contained 500 mg SOCE, while the tablet for the placebo group contained cellulose crystals and had the same appearance, weight, and characteristics as the test products (Table 1).

### 2.4. Outcome Measurements

#### 2.4.1. Measurement of Cognitive Function

Computerized Neurological Function Test (CNT) version 4.0 (MAXMEDICA Inc., Seoul, Korea) was used to evaluate adult neurocognitive function [23,33,34]. Subjects underwent this test before baseline (week 0) and after 12 weeks of the study.

#### 2.4.2. Primary Outcomes

The primary output measures of this study were changes in memory function tests. Specifically, among the 18 sub-categories of the CNT, the visual learning test (Visual LT) and verbal learning test (Verbal LT) were considered as the appropriate tests for evaluation of cognitive function in this study [34]. As memory deficits are major symptoms of MCI and attention is a prerequisite for memory encoding, decline in these cognitive functions are commonly reported in MCI [35]. Therefore, we selected the most appropriate subtests to examine these cognitive domains of older adults with memory impairment.

CNT responded by pressing the monitor by hand or pressing a button device while the study subject was looking at the touch screen monitor, and the CNT tester performed it based on standardized phrases. Visual LT is a test that shows 15 figures on a computer monitor and then memorizes the first figure out of a total of 30 figures. Visual LT was measured by repeating a total of 5 times (A1 to A5). Visual LT recognition was measured by showing a total of 30 figures on a monitor to the subject after 20 min of Visual LT was implemented and then reminding the figures shown earlier.

In the verbal LT, the subjects listened to a list of 15 words (A list) five times (Trial A1~A5) via the recorded voice of the computer speaker and recalled out loud as many words as possible after each trial (immediate recall). After the fifth trial (A5), a new interference list (list B) was presented and recalled. After presenting list B (Verbal LT B), the subjects recalled the words from list A again (Verbal LT A6). After a 20-min delay, subjects were asked to recall as many original words from list A as possible (Verbal LT A20 delayed recall). They were then asked to select words from list A out of 50 words presented on the computer screen containing 15 words from list A (Verbal LT REC delay recognition).

Five scores were estimated as outcome measures in this study: (1) the total number of words recalled immediately after trials A1 to A5 (A1+A2+A3+A4+A5; Verbal LT A1A5 total learning index), (2) the difference between A5 and A1 (A5−A1; Verbal LT learning Slope A5−A1), (3) the total number of words recalled after a 20-min delay (Verbal LT A20 delayed recall), (4) the total number of words correctly selected from the 50-word list (Verbal LT REC delay recognition), and (5) the difference between A5 and A20 delayed recall (A5-A20; Verbal LT A5-A20 memory retention). Visual working memory test (Visual WMT) was conducted. Visual WMT was measured by making the subjects give a total of 15 words with the same frequency of use through a computer speaker and having them speak as they remember them, regardless of the order.

#### 2.4.3. Secondary Outcomes

Measurement of amyloid-β in plasma:

Blood (3 mL) for plasma amyloid-β (Aβ) assessment was collected in an EDTA tube and blood was mixed by inverting the tube 10 times right after sampling. Samples were centrifuged at 3000 rpm for 10 min, and 1 mL of the supernatant was transferred into a 1.5 mL micro tube. The sample was then immediately stored in a freezer (−70 °C). Aβ 40 and Aβ 42 protein levels in the plasma samples were measured using an enzyme-linked immunosorbent assay (ELISA) using Human Amyloid Beta Assay Kit (Immuno-Biological Laboratories, Takasaki-shi, Japan) [36], which is a solid-phase sandwich ELISA assay.

Measurement of 8-OHdG (8-hydroxy-2′-deoxyguanosine) level in urine:

The concentration of 8-OHdG (8-hydroxy-2′-deoxyguanosine) is a quantitative marker of oxidative damage to DNA. For the 8-OHdG analysis, urine samples were collected from all subjects after fasting for at least 12 h. Approximately 1 mL urine was transferred to a 1.5 mL micro tube without treatment and immediately stored in a refrigerator (−70 °C) for further analysis. 8-OHdG was measured from stored urine using the BIOXYTECH^®^ 8-OHdG EIA kit (OXIS Health Products, Inc., Portland, OR, USA) following the manufacturer’s instructions [37].

### 2.5. Safety Measurements

To investigate the adverse effects, the clinical condition of the subjects and their vital signs were assessed before baseline (week 0) prior to the study participation and after 12 weeks of study participation. A safety assessment, including general blood and chemical tests, was also performed. Blood samples were collected after 12 h of fasting and centrifuged (Hanil Science Industrial Co. Ltd., Seoul, Korea) at 3000 rpm for 20 min to separate plasma. Plasma was stored at −80 °C until analysis. The hematological parameters such as WBC, RBC, hemoglobin, hematocrit, and platelet count were analyzed. Biochemical tests were conducted to assess total bilirubin, ALP, gamma-GT, ALT, AST, total cholesterol, triglycerides, HDL-C, LDL-C, glucose, total protein, albumin, BUN, creatinine, creatine kinase, and LDH in blood. Urine samples were tested for specific gravity, pH, WBC, nitrite, protein, glucose, ketones, urobilinogen, bilirubin, and occult blood. All biochemical analyses were performed by the clinical pathology department of our hospital.

### 2.6. Investigation of Dietary Intake and Physical Activity

Dietary intakes of the participants were assessed by diet records at the first visit, baseline (week 0), and after 12 weeks of participation in the study. All dietary intakes for 3 days (two days of the week, one day of the weekend) before the baseline visit and for the third visit (week 12) were recorded, and retrieved diet records were analyzed. Daily average intake was calculated using Can-Pro 4.0 software (The Korean Nutrition Society, Seoul, Korea) with the data from the 3 days recorded during 12-week study from which the average daily calorie and nutrition intake were calculated. Physical activity was evaluated according to a metabolic equivalent task (MET) assessment using the global physical activity questionnaire (GPAQ) [38]. The MET value was used to analyze the physical activity or GPAQ data, representing the relative proportion of a subject’s working metabolic rate to resting metabolic rate.

### 2.7. Sample Size

The sample size calculation for this study was based on the assumption that the variation in the verbal learning memory test after 12 weeks of intake in the SOCE group would be +1.59, the variation in the placebo group would be −1.43, and the standard deviation would be 3.71. The number of subjects required was calculated as described by Chung et al. (2014) [39]. The number of subjects needed in each group to achieve 80% power for a 5% significance level with a two-sided test was 25 persons per group. A total of 70 subjects (35 subjects per group) was required assuming a dropout ratio of 30%. Therefore 70 subjects were enrolled for 1:1 randomization to the SOCE and placebo groups.

### 2.8. Statistical Analysis

All statistical processing was performed using SAS version 9.2 (SAS Institute, Cary, NC, USA) and SPSS version 20 (IBM Co., Armonk, NY, USA). Continuous variables were presented as means ± SD, and categorical variables are presented as frequencies. The significance of differences in categorical variables was assessed by Chi-square tests (Fisher’s exact test), and comparison of means between the two groups was performed using the independent *t*-test. Efficacy test variables in this study were analyzed in the intent-to-treat (ITT) group. Changes in the visual memory test, visual cognition working memory test, and verbal memory test scores were analyzed using the paired *t*-test before and after 12 weeks of the test/control product intake. Changes in measurements over the 12-week study period were obtained by calculating the difference between each group’s pre- and post-intervention measurements. Analysis was implemented using the independent t-test. Analysis of covariance (ANCOVA) adjusted for visual working memory items was used to determine whether the observed difference in verbal learning between the two groups was independent of visual working memory items. As an effective secondary measure, differences in serum β-amyloid and 8-OHdG in urine in each intake group before and after 12 weeks of intake were analyzed using the paired t-test; analysis was implemented using the independent *t*-test.

According to a meta-analysis study [40], positive associations were found among white matter hyperintensity volume, depression, diabetes, hypertension, old age, and female gender of risk factors for predicting progression from MCI to AD. Accordingly, subjects with stage 2 hypertension (systolic blood pressure of 160 mmHg or diastolic blood pressure of 90 mmHg), which can be classified as severe hypertension, are excluded from those who have taken drugs for the treatment of hyperthyroidism or hypothyroidism [41]. After that, subgroup analysis was performed to analyze and compare Z scores. In addition, the Z score was calculated using three tests reflecting the cognitive function (Visual LT, Verbal LT, and Visual WMT). The sign of the Z-score is that the positive Z score is better performance than the average score of the placebo group. The standardized Z score of the three tests was calculated and averaged to obtain one composite score for memory. The composite score and combined composite score of visual memory, verbal learning memory, and working memory between the two groups were analyzed using independent t-test. The baseline value of each efficacy evaluation item and the non-homogeneous demographic information items were corrected by covariate and ANCOVA was performed. The statistical significance was set at *p* < 0.05.

## 3. Results

### 3.1. Characteristics of the Participants

A total of 80 subjects who volunteered to participate in this clinical study were selected and participated in the screening test, from which a total of 70 were selected based on our inclusion and exclusion criteria. The general characteristics of the subjects are shown in Table 2. The proportion of male and female subjects in the study was 35.7% and 64.3%, respectively, and the average age was 69.9 years. There were no statistically significant differences in sex distribution, age, drinking, smoking, blood pressure, physical measurement indices, or cognitive function scores between the two groups. However, there was a statistically significant difference in Visual WMT (correct response) between the groups, with increased Visual WMT (correct response) in the SOCE group and decreased in the placebo group (*p* < 0.05) data not shown. Three of the 70 selected subjects withdrew their consent, seven failed to comply with the test products, and two left the study due to severe adverse events (SAEs). A total of 70 subjects completed all the procedures specified in the human study plan. Data from 70 subjects were used for analyses (Figure 2).

### 3.2. Adherence to Treatment

Adherence (%) was measured by calculating the percentage of returned test products to the percentage of the ingested product. This was only calculated for those participants who completed the study. Compliance rates of the SOCE and placebo groups were 85.1% and 89.7%, respectively. The difference between these two groups in adherence was not significant (*p* > 0.05). Data not shown.

### 3.3. Primary Outcome Measure

In this study, changes in the cognitive function scores in the SOCE group after 12 weeks of intake were significantly greater than those in the placebo group (Table 3). Verbal LT A2 and Verbal LT A3 scores in the SOCE group were significantly better after 12 weeks of intake than before the intake (*p* = 0.004 and *p* = 0.000, respectively). There was a statistically significant difference trend between these two groups (*p* > 0.05). Verbal LT A6 score in the SOCE group was significantly better after 12 weeks of intake than prior to the intake (*p* = 0.001). While there was no statistically significant difference in Verbal LT A6 score between the two groups (*p* = 0.082), there was a statistically significant difference in this score between these two groups when corrected for baseline (*p* = 0.046). There was a statistically significant increase in the verbal LT delayed recognition item score in the SOCE group compared to the placebo group after correction for baseline (*p* = 0.033). Verbal LT score was significantly higher after 12 weeks of intake than before intake in the SOCE group (*p* = 0.008). Furthermore, there was a statistically significant difference in the verbal LT score between the SOCE and placebo groups after 12 weeks (*p* = 0.020). The accuracy of the visual WMT was significantly improved after 12 weeks of intake compared to prior to intake in the placebo group (*p* = 0.013), but on baseline adjusting, there was no significant difference in the visual WMT scores between the two groups (*p* > 0.05).

In this study, the results of the change in the Z-score for each memory function item were presented in Table 4 by sub-analysis excluding those who took drugs for metabolic disease treatment (blood pressure, thyroid function). After taking SOCE for 12 weeks, the Z-score of recognition and the composite Z-score of the verbal memory domain of the verbal memory function significantly increased compared to the placebo group (*p* < 0.05). Further, global cognitive function change was analyzed using the average value of the overall domain composite scores of Visual LT, Visual WMT, and Verbal LT, where SOCE group showed significant improvement in memory compared to the placebo group (z = 2.07, *p* < 0.05).

### 3.4. Secondary Outcomes

#### 3.4.1. Changes in Plasma Amyloid-β Levels in Blood and 8-OHdG Level in Urine

Changes in plasma amyloid-β and 8-OHdG levels of the subjects are presented in Table 5. Plasma Aβ(1-40) and Aβ(1-42) levels were significantly lower in the SOCE group than in the placebo group (*p* = 0.041, *p* = 0.021). The level of 8-OHdG, which evaluates the degree of damage to DNA, was significantly decreased in the placebo group after 12 weeks of intake compared to prior to intake. However, there was no statistically significant difference between the SOCE and placebo groups (*p* > 0.05).

#### 3.4.2. Associations between Changes in Plasma Amyloid-β Levels and Cognitive Performance

Here we investigated the associations between the combined composite score and plasma amyloid-β levels. In this study, the correlation between Aβ 42/40 ratio and combined composite score revealed a negative correlation in the placebo group (r = −0.547, *p* = 0.015). There was no significant association between change in plasma Aβ 42/40 ratio and cognitive performance in the SOCE group (r = 0.067, *p* = 0.768) (Appendix A).

#### 3.4.3. Dietary Intake and Physical Activity

ITT was analyzed to measure the dietary intake and observed no statistically significant difference between SOCE and placebo groups (Appendix A). Similarly, there was no statistically significant difference in physical activity (MET value) between groups (*p* > 0.05). Additionally, there was no statistically significant difference in physical activity (MET value) between groups (*p* > 0.05).

### 3.5. Safety and Tolerability

During the period of participation in this study, seven subjects (two from SOCE group and five from placebo group) showed abnormal reactions (*p* > 0.05). Five of the seven adverse cases were mild adverse reactions, and two were severe adverse reactions. However, there was no major difference in the occurrence of adverse reactions between these two groups, and adverse reactions were either unrelated or highly unlikely due to the intake of the test products. Vital signs and biochemical markers (hematological test, blood biochemistry test, and urine test) were determined to be within reference ranges and thus had no clinical significance (Appendix A).

## 4. Discussion

This study is the first randomized clinical trial (RCT) to evaluate the efficacy and safety of SOCE supplementation to prevent MCI in older adults with memory impairment. A dose of 1.5 g of SOCE three times a day improved the verbal memory in the SOCE group than the placebo group and significantly reduced the levels of blood amyloid-β fragments in SOCE group. The absence of adverse reaction confirms the safety of SOCE supplementation during the study. These findings are consistent with those of in vitro and animal studies [30,31,32], which showed that SOCE improved memory function in animal models. Previously, three different investigations by Um et al. evaluated the effect of SG in improving memory function. Firstly, they demonstrated that SG inhibited the cognitive deficits caused by aging through its antioxidant activity in senescence-accelerated mice P8 (SAMP8) [30]. Later, they showed that SG has a protective effect against neuronal apoptosis caused by amyloid β25–35 by scavenging oxidative radicals and regulating MAPK signaling pathways in SK N SH cells [31]. Finally, Um et al. [32] confirmed that SG supplementation protected from cognitive deficits induced by Aβ through its antioxidant activity in mice treated by internal brain injections of Aβ protein, which is a causative agent of dementia. These observations suggest that the SG in SOCE protects against neuronal apoptosis caused by Aβ. Hence, SOCE supplementation protects nerve cells via scavenging of oxidative radicals and altered MAPK signaling [32]. In this study, CNTs detected mild degrees of cognitive impairment and can be used efficiently to assess treatment efficiency against cognitive impairment. Among CNT’s, the Verbal LT is a clinical assessment of verbal learning and memory that represents relevance to measuring the effects of numerous neurological conditions, including MCI. Here, the Verbal LT score in the SOCE group showed significantly higher than the placebo group.

In Verbal LT, the SOCE group confirmed the increase in their Verbal LT A6 index after 12 weeks of SOCE intake. A higher Verbal LT A6 index indicates the recall of learned information (List A) is less disturbed by other similar information (List B) presented thereafter. During Verbal LT in the SOCE group, the short-term store competes with new information (List B) unrelated to old information (List A). This process is necessary for successful memory retrieval and improves cognitive control. In this study, supplementation of SOCE potentially improved memory retrieval. Moreover, subjects in the SOCE group showed improvement in the verbal learning delayed recognition test (Verbal LT REC delay recognition index) than the placebo group. These observations are similar to previous reports where cognitive control is closely associated with the process underlying recognition [42]. In addition, cognitive Z-score change sub-analysis, the composite Z-score and the global domain composite score of the verbal memory domain were significantly increased in the SOCE group than the placebo group indicating cognitive control. These outcomes suggest that memory can be improved as it is linked to long-term memory through the learning effect of repetition of the immediate recall test.

In general, aggregation and accumulation of Aβ proteins in the brain have been considered a defining pathology associated with AD, and an increase in plasma amyloid β levels is closely related to the development of AD [38]. The observations of previous investigations suggest strong correlations between the plasma Aβ and areas of high Aβ deposition in the brain [41,43,44]. Moreover, plasma Aβ42/Aβ40 ratio is a promising AD biomarker that predicts cerebral amyloid and AD pathology in at-risk individuals [45].

However, the reported results concerning Aβ are contradictory. Several studies reported that a high Aβ 40, and low Aβ 42 levels, and a greater increase in Aβ 42/40 ratio are associated with the risk of developing AD [46,47,48,49]. At the same time, several extensive studies have consistently reported that a lower Aβ 42/40 ratio in plasma is associated with higher risk of AD [50]. Recently, a study by Boada et al. reported [51] plasma exchange (PE) with albumin replacement with human albumin modified cerebrospinal fluid (CSF) and plasma Aβ (1–42) levels. AD subjects treated with PE showed improvement in memory and language functions. The observations of this study were similar, and the significant decreases in Aβ (1–40) and Aβ (1–42) levels in the SOCE group after 12 weeks of intake compared to before intake suggests that intake of SOCE protected against a decline in memory by reducing the concentration of Aβ. Additionally, there were no significant differences in calorie intake, nutrient intake, or the extent of physical activity between the two groups. Study participants maintained their routine activities during their participation in this study, and thus dietary intake and physical activity will not influence our findings during the study period. This study is the first of its kind as improvements in memory function in humans with supplementation of SOCE was clinically evaluated. Moreover, SOCE, which is used as feedstock or compost, was identified as a functional material for treating older adults with memory impairment. The study has some limitations. The results of this study cannot be generalizable due to the small number of participants and it does not assess attention, which is the core ability for encoding information. However, the results of this study are consistent with those of previous studies that reported supplementation with SOCE can improve cognitive function and memory [30,31,32]. Besides, Aβ levels are precisely measured using positron-emission tomography (PET), which is an invasive method. Current observations with respect to blood Aβ concentrations may not accurately reflect the intensity of the disease. Our results indicate the validity and safety of 12-week supplementation with SOCE. Thus, it is necessary to conduct extensive investigations with a significant number of volunteers to address the limitations. However, key observations from this study suggest that SOCE supplementation in older adults with memory impairment is safe and may help to prevent and manage AD.

## 5. Conclusions

In summary, 12-week SOCE supplementation in older adults with memory impairment resulted in improved global cognitive function than the control group, particularly improved verbal learning memory function.

## Figures and Tables

**Figure 1 nutrients-13-02606-f001:**
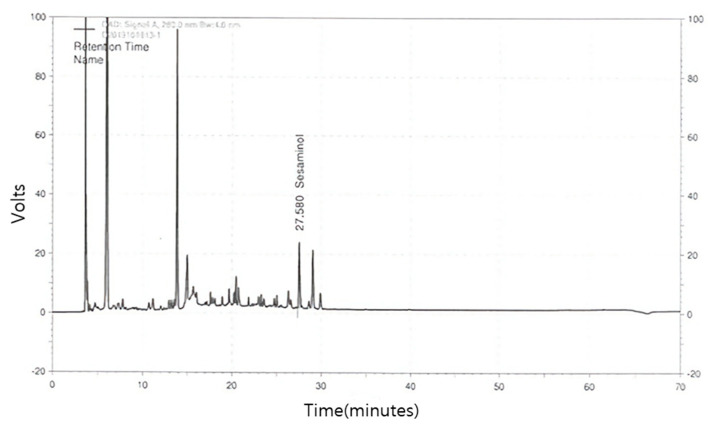
Representative chromatograms of SOCE (sesame oil cake extract) based on high-performance liquid chromatography (HPLC) analysis of sesaminol.

**Figure 2 nutrients-13-02606-f002:**
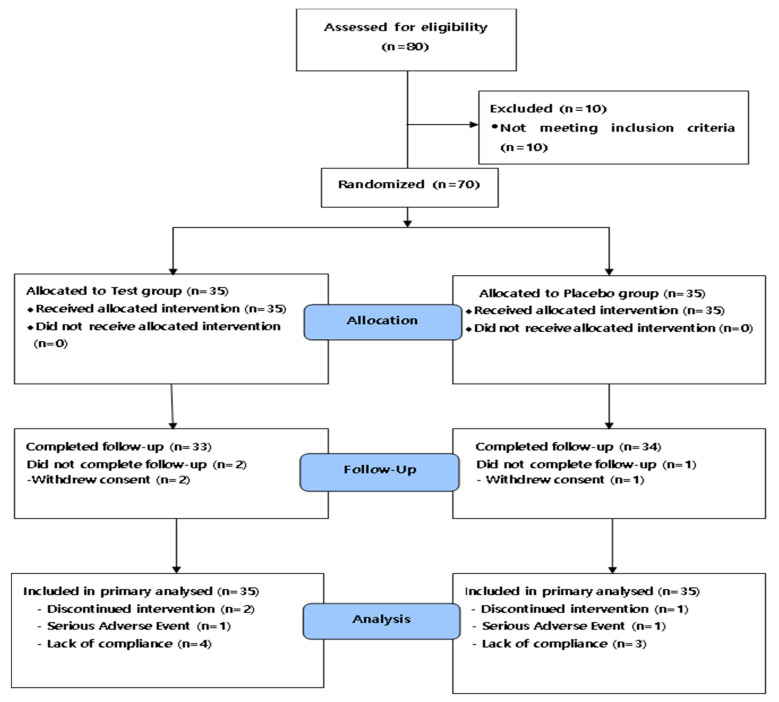
Flow diagram showing the selection and allocation of participants in the study.

**Table 1 nutrients-13-02606-t001:** Composition of the test products.

Component	Test SOCE Supplement (%)	Placebo Supplement (%)
SOC extract	72	-
Cellulose crystal	19	80.2
HPMC	4.2	2
Magnesium stearate	2.8	1.3
Silicon dioxide	2	1.5
Gardenia yellow	-	8.2
Caramel coloring	-	6.8
Food coloring, Yellow #5	-	0.03
Total	100	100

SOCE, sesame oil cake extract; SOC, sesame oil cake; HPMC, hydroxypropyl methylcellulose.

**Table 2 nutrients-13-02606-t002:** Demographic characteristic of study participants.

	SOCE Group (*n* = 35)	Placebo Group (*n* = 35)	Total (*n* = 70)	*p*-Value
Sex (male/female)%	9(25.7)/26(74.3)	16(45.7)/19(54.3)	25(35.7)/45(64.3)	0.081 ^(1)^
Academic ability (years)	10.14 ± 3.78	9.29 ± 3.88	9.71 ± 3.83	0.353 ^(2)^
Age (years)	68.69 ± 5.33	71.14 ± 5.62	69.91 ± 5.58	0.065 ^(2)^
Drink alcohol (yes/no) %	6(17.1)/29(82.9)	4(11.4)/31(88.6)	10(14.3)/60(85.7)	0.495 ^(1)^
Alcohol amount consumed (units/week)	0.21 ± 0.64	0.23 ± 0.89	0.22 ± 0.77	0.902 ^(2)^
Smoker (yes/no) %	2(5.7)/33(94.3)	2(5.7)/33(94.3)	4(5.7)/66(94.3)	1.000 ^(1)^
No. cigarettes per day	0.29 ± 1.18	0.11 ± 0.53	0.20 ± 0.91	0.435 ^(2)^
SBP (mmHg)	131.69 ± 14.10	133.89 ± 11.79	132.79 ± 12.95	0.481 ^(2)^
DBP (mmHg)	77.29 ± 9.31	79.17 ± 10.19	78.23 ± 9.74	0.422 ^(2)^
Pulse (beats/minute)	70.00 ± 9.32	71.03 ± 9.49	70.51 ± 9.35	0.649 ^(2)^
Height (cm)	155.86 ± 7.24	157.51 ± 8.53	156.69 ± 7.90	0.384 ^(2)^
Weight (kg)	61.96 ± 7.98	62.43 ± 8.41	62.19 ± 8.14	0.813 ^(2)^
Body mass index (kg/m^2^)	25.50 ± 2.62	25.17 ± 2.78	25.33 ± 2.69	0.616 ^(2)^
CERAD-K word list memory	13.37 ± 2.66	13.71 ± 3.06	13.54 ± 2.85	0.619 ^(2)^
CERAD-K word list recall	3.97 ± 1.25	3.60 ± 1.50	3.79 ± 1.38	0.264 ^(2)^
CERAD-K word list recognition	7.63 ± 1.65	7.09 ± 1.92	7.36 ± 1.79	0.208 ^(2)^

Values are presented as means ± SDs or numbers (percentages). ^(1)^ Analyzed by chi-square test. ^(2)^ Analyzed by independent *t*-test Abbreviation: SOCE, sesame oil cake extract; CERAD-K, Korean version of the Consortium to Establish a Registry for Alzheimer’s Disease Assessment Packet.

**Table 3 nutrients-13-02606-t003:** Changes in neurocognitive function tests before and after 12 weeks of intake.

	SOCE Group (*n* = 35)				Placebo Group (*n* = 35)					
	Baseline	12 Week	Change Value	*p* ^(1)^	Baseline	12 Week	Change Value	*p* ^(1)^	*p* ^(2)^	*p* ^(3)^
Visual LT A1	8.15 ± 2.46	9.18 ± 1.78	1.03 ± 2.21	0.012	8.16 ± 2.37	10.00 ± 1.55	1.84 ± 2.13	0.000	0.136	0.029
Visual LT A2	10.09 ± 2.10	10.09 ± 2.11	0.00 ± 2.06	1.000	9.77 ± 1.93	10.31 ± 1.39	0.54 ± 2.09	0.134	0.285	0.400
Visual LT A3	10.21 ± 2.06	10.76 ± 1.85	0.55 ± 2.39	0.198	10.69 ± 1.51	10.97 ± 1.26	0.28 ± 1.84	0.393	0.619	0.759
Visual LT A4	10.84 ± 2.13	11.06 ± 2.17	0.22 ± 2.43	0.615	10.61 ± 1.41	11.12 ± 1.19	0.52 ± 1.46	0.051	0.552	0.728
Visual LT A5	11.16 ± 1.97	11.63 ± 1.70	0.47 ± 1.83	0.158	10.74 ± 1.88	11.23 ± 1.48	0.49 ± 1.77	0.114	0.969	0.505
Visual LT (recognition)	10.85 ± 2.00	11.06 ± 1.92	0.21 ± 2.25	0.591	11.09 ± 1.51	10.75 ± 1.63	−0.34 ± 1.70	0.260	0.266	0.336
Visual WMT (accuracy)	36.57 ± 19.34	34.70 ± 16.17	−1.87 ± 13.34	0.419	27.29 ± 15.98	33.86 ± 15.14	6.57 ± 13.62	0.010	0.013	0.111
Visual WMT (reaction time)	622.02 ± 89.32	602.16 ± 77.42	−19.86 ± 75.66	0.148	656.22 ± 81.81	627.47 ± 75.08	−28.75 ± 101.58	0.114	0.691	0.456
Verbal LT A1	5.03 ± 1.58	5.35 ± 1.47	0.32 ± 1.45	0.224	4.70 ± 1.37	5.87 ± 1.61	1.17 ± 1.91	0.002	0.056	0.090
Verbal LT A2	7.22 ± 1.54	8.19 ± 1.89	0.97 ± 1.77	0.004	7.25 ± 1.74	7.41 ± 1.81	0.16 ± 1.97	0.657	0.087	0.063
Verbal LT A3	7.79 ± 1.87	9.15 ± 2.00	1.36 ± 2.00	0.000	8.13 ± 1.86	8.72 ± 2.04	0.59 ± 1.54	0.037	0.087	0.124
Verbal LT A4	8.48 ± 2.17	10.00 ± 1.80	1.52 ± 1.91	0.000	8.97 ± 2.19	9.79 ± 2.25	0.82 ± 2.24	0.039	0.178	0.313
Verbal LT A5	9.31 ± 1.80	10.13 ± 1.56	0.81 ± 1.62	0.008	9.24 ± 2.18	9.91 ± 2.44	0.68 ± 2.07	0.065	0.768	0.690
Verbal LT B	4.42 ± 1.39	4.48 ± 1.20	0.06 ± 1.52	0.820	4.42 ± 1.31	4.45 ± 0.93	0.03 ± 1.40	0.899	0.939	0.902
Verbal LT A6	7.44 ± 2.20	8.56 ± 2.40	1.13 ± 1.79	0.001	7.30 ± 2.24	7.52 ± 2.28	0.21 ± 2.33	0.604	0.082	0.046
Verbal LT A20 ^(4)^ (delayed recall)	6.38 ± 2.23	8.13 ± 2.32	1.75 ± 2.42	0.000	6.64 ± 2.78	7.85 ± 3.02	1.21 ± 2.30	0.005	0.362	0.420
Verbal LT REC (delay recognition) ^(5)^	11.94 ± 1.81	12.48 ± 1.57	0.55 ± 2.03	0.143	10.88 ± 2.45	11.27 ± 1.89	0.39 ± 2.52	0.377	0.789	0.033
Verbal LT A1A5 (Total) ^(6)^	38.84 ± 7.02	42.94 ± 6.18	4.09 ± 5.90	0.000	39.06 ± 8.23	41.88 ± 8.99	2.82 ± 7.03	0.025	0.431	0.420
Verbal LT A1A5 (average)	7.77 ± 1.40	8.59 ± 1.24	0.82 ± 1.18	0.000	7.81 ± 1.65	8.38 ± 1.80	0.56 ± 1.41	0.025	0.431	0.420
Verbal LT (Learning Slope A5-A1) ^(7)^	3.53 ± 1.72	4.41 ± 1.76	0.88 ± 1.76	0.008	4.12 ± 1.85	3.82 ± 1.78	−0.30 ± 2.20	0.435	0.020	0.058
Verbal LT A5-A20 Memory retention ^(8)^	3.53 ± 172	1.94 ± 1.64	−0.70 ± 2.38	0.102	2.48 ± 1.65	2.23 ± 1.75	−0.26 ± 2.38	0.551	0.464	0.482

Values are presented as means ± SD ^(1)^ Change between baseline and 12 weeks analyzed by paired *t*-test. ^(2)^ Change value compared between groups by independent *t*-test. ^(3)^ ANCOVA (baseline values included as covariates; visual working memory test *p* = 0.038). Memory function scores were estimated as outcome measures in this study: ^(4)^ The total number of words recalled after a 20-min delay (Verbal LT A20 delayed recall). ^(5)^ The total number of words correctly selected from the 50-word list (Verbal LT REC delay recognition). ^(6)^ The total number of words recalled immediately after trials A1 to A5 (A1+A2+A3+A4+A5; Verbal LT A1A5 total learning index). ^(7)^ The difference between A5 and A1 (A5-A1; Verbal LT learning Slope A5-A1). ^(8)^ The difference between A5 and A20 delayed recall [A5-A20; Verbal LT A5−A20 memory retention]. Abbreviation: SOCE, sesame oil cake extract; Visual LT, visual learning test; Visual WMT, Visual working memory test; Verbal LT, verbal learning test.

**Table 4 nutrients-13-02606-t004:** Differences in composite Z-scores of each cognitive domain and combined cognitive function between the placebo and SOCE groups.

	SOCE Group	Placebo Group	Z ^(3)^	*p*-Value ^(1)^
	12 Week ^(2)^	12 Week ^(2)^		
Visual memory function †				
Immediate recall	0.38 ± 1.10	0.41 ± 0.54	−0.13	0.900
Recognition	0.12 ± 1.02	−0.20 ± 0.79	1.23	0.221
Domain composite score	0.25 ± 0.98	0.10 ± 0.53	0.70	0.483
Verbal memory function ‡				
Immediate recall	0.63 ± 0.85	0.18 ± 0.95	1.77	0.076
Delayed recall	0.84 ± 0.80	0.54 ± 0.96	1.23	0.218
Recognition	0.65 ± 0.57	0.04 ± 0.72	3.22	0.001
Domain composite score	0.73 ± 0.59	0.25 ± 0.76	2.43	0.015
Working memory function ^§^				
Visual working memory test (adjusted accuracy)	0.34 ± 0.70	0.11 ± 0.82	1.13	0.260
Visual working memory test Reaction time	−0.59 ± 0.92	−0.46 ± 1.06	−0.48	0.635
Domain composite score	−0.09 ± 0.61	−0.18 ± 0.77	0.45	0.654
Combined cognitive function ^∥^				
Domain composite score	0.33 ± 0.49	0.03 ± 0.44	2.07	0.039

Values are presented as z score ± SD for the SOCE (*n* = 23) and Placebo (*n* = 29) groups after, are excluded from the severe hypertension subjects and those who have taken drugs for the treatment of hyperthyroidism or hypothyroidism. ^(1)^ Analyzed by independent *t*-test. ^(2)^ Change in a Z score from baseline to 12 weeks. ^(3)^ Analyzed by Z-test between groups. †; Visual memory function: Immediate recall (Visual learning test A1~A5) + Recognition. ‡; Verbal memory function: Immediate recall (Verbal learning test A1~A5) + Delayed recall (Verbal learning test A20) + Recognition (Verbal learning test REC). §; Working memory function: Visual working memory test (adjusted accuracy) + Visual working memory function reaction time. ∥; Combined cognitive function: Visual memory function + Verbal memory function + Working memory function.

**Table 5 nutrients-13-02606-t005:** Changes in amyloid β and 8-OHdG levels before and after 12 weeks of intake of study products.

		SOCE Group (*n* = 35)			Placebo Group (*n* = 35)				
	Baseline	12 Week	Change Value	*p* ^(1)^	Baseline	12 Week	Change Value	*p* ^(1)^	*p* ^(2)^
Amyloid β (1–40) (pg/mL)	298.93 ± 53.11	283.88 ± 52.81	−15.05 ± 41.71	0.054	307.35 ± 53.51	312.76 ± 61.67	5.41 ± 35.76	0.399	0.041
Amyloid-β (1–42) (pg/mL)	1.74 ± 0.44	1.61 ± 0.17	−0.13 ± 0.35	0.043	1.61 ± 0.17	1.69 ± 0.41	0.08 ± 0.35	0.225	0.021
Amyloid-β (42/40)	0.01 ± 0.00	0.01 ± 0.00	0.00 ± 0.00	0.901	0.01 ± 0.00	0.01 ± 0.00	0.00 ± 0.00	0.310	0.319
8-OHdG (ng/mg creatinine)	9.63 ± 1.87	9.13 ± 2.36	−0.50 ± 2.29	0.227	9.17 ± 1.90	8.55 ± 2.13	−0.62 ± 1.53	0.029	0.805

Values are presented as means ± SD. ^(1)^ Change between baseline and 12 weeks analyzed by paired *t*-test. ^(2)^ Change value compared between groups by independent *t*-test. Abbreviation: SOCE, sesame oil cake extract.

## Data Availability

The datasets generated during and/or analyzed during the current study are available from the corresponding author on reasonable request.

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
