# Peer review of "Efficacy and Safety of Sesame Oil Cake Extract on Memory Function Improvement: A 12-Week, Randomized, Double-Blind, Placebo-Controlled Pilot Study"

_nutrients, 2021, doi:10.3390/nu13082606_

Round 1
Reviewer 1 Report
I appreciate to have a chance to review a nice study. The study demonstrated the effects of sesame oil cake extract on the cognitive function of MCI subjects. It seems to be performed in a appropriate fashion, and the manuscript was well-written.
I only have several minor comments.
- P1 line 39 Alzheimer’s disease.
Recently “Alzheimer’s disease” sometimes includes preclinical and prodromal stage of the disease, not necessarily means dementia. The authors may want use, foe ex. Dementia of Alzheimer type, or Alzheimer dementia etc.
- P1 line 44-45
Please recheck the sentence beginning with “Since AD is----
- P 3 line 134
Please recheck the sentence beginning with “ Sesame oil a byuproduct----
- P4 line 148
Please recheck the sentence beginning with “If acceptable -----
- P2 line 91
“Memory index socres below 1.0 and 2.0 SD “
I do not understand this, between 1.0 and 2.0 SD?
- P2 line 76
Older adults with “subjective” memory complaints.
“Subjective memory co
I appreciate to have a chance to review a nice study. The study demonstrated the effects of sesame oil cake extract on the cognitive function of MCI subjects. It seems to be performed in a appropriate fashion, and the manuscript was well-written.
I only have several minor comments.
- P1 line 39 Alzheimer’s disease.
Recently “Alzheimer’s disease” sometimes includes preclinical and prodromal stage of the disease, not necessarily means dementia. The authors may want use, foe ex. Dementia of Alzheimer type, or Alzheimer dementia etc.
- P1 line 44-45
Please recheck the sentence beginning with “Since AD is----
- P 3 line 134
Please recheck the sentence beginning with “ Sesame oil a byuproduct----
- P4 line 148
Please recheck the sentence beginning with “If acceptable -----
- P2 line 91
“Memory index socres below 1.0 and 2.0 SD “
I do not understand this, between 1.0 and 2.0 SD?
- P2 line 76
Older adults with “subjective” memory complaints.
“Subjective memory complaints” sometimes imply the lack of objective memory decline. If it is not the case, please reword these terms thoroughly.
Author Response
Reviewer 1
I appreciate to have a chance to review a nice study. The study demonstrated the effects of sesame oil cake extract on the cognitive function of MCI subjects. It seems to be performed in a appropriate fashion, and the manuscript was well-written.
I only have several minor comments.
Q1. P1 line 39 Alzheimer’s disease.
Recently “Alzheimer’s disease” sometimes includes preclinical and prodromal stage of the disease, not necessarily means dementia. The authors may want use, foe ex. Dementia of Alzheimer type, or Alzheimer dementia etc.
Response: We thank reviewer for the valuable comment. Agreeing with the reviewer’s suggestion, we have replaced Alzheimer’s disease with dementia of Alzheimer’s type (DAT)
Q2. P1 line 44-45
Please recheck the sentence beginning with “Since AD is----
Response: As suggested, we have corrected the sentence. The updated sentence is as follows,
“Neuropsychological symptoms and pathological changes accompany AD, and no effective treatment option is available”
Q3. P 3 line 134
Please recheck the sentence beginning with “ Sesame oil a byuproduct----
Response: As suggested, we have modified the sentence. The updated sentence is as follows,
“Sesame oil cake (SOC), a byproduct of the sesame oil extraction process from sesame seeds”.
Q4. P4 line 148
Please recheck the sentence beginning with “If acceptable -----
Response: We have verified the sentence and found that it is unnecessary in that part of the method. Thus, it is deleted.
Q5. P2 line 91
“Memory index scores below 1.0 and 2.0 SD “
I do not understand this, between 1.0 and 2.0 SD?
Response: We apologize for the error. We have rectified the sentence in the revised version. The updated sentence is as follows,
“Subjects memory index scores that fell greater than 1 standard deviations (SDs) from the normal mean value for each test item in the neuropsychological part of (word list memory, word list recall, and word list recognition test) the Korean version of the Consortium to Establish a Registry for Alzheimer’s Disease Assessment Packet (CERAD-K) (1)”.
References
- Lee J. H., Lee K. U., Lee D. Y., Kim K. W., Jhoo J. H., Kim J. H., et al. (2002). Development of the Korean version of the Consortium to Establish a Registry for Alzheimer’s Disease Assessment Packet (CERAD-K): clinical and neuropsychological assessment batteries. J. Gerontol. B Psychol. Sci. Soc. Sci. 57 47–53.
A Here is the CERAD-K assessment of normative table (For example)
Q6. P2 line 76
Older adults with “subjective” memory complaints. “Subjective memory complaints” sometimes imply the lack of objective memory decline. If it is not the case, please reword these terms thoroughly.
Response: We thank reviewer for the comment. As suggested, we have replaced these words with older adults (aged above 60 years) with memory impairment. The updated sentence is as follows,
“This study determines the effects of SOCE on cognitive function in older adults (aged above 60 years) with memory impairment”
Thank you!

Reviewer 2 Report
This is an interesting study that determined the effects of SOCE on cognitive function in 75 older adults (over 60 years of age) with subjective memory impairment. The introduction is adequate. The clinical trial was well conducted. I recommend checking the tables. As for the number of the sample size enrolled, from my calculations it is underestimated. As the authors say this is a pilot study, should you specify this in the title.
Author Response
Reviewer 2
Comments and Suggestions for Authors
Q1. This is an interesting study that determined the effects of SOCE on cognitive function in 75 older adults (over 60 years of age) with subjective memory impairment. The introduction is adequate. The clinical trial was well conducted. I recommend checking the tables. As for the number of the sample size enrolled, from my calculations it is underestimated. As the authors say this is a pilot study, should you specify this in the title.
Response: We appreciate reviewers’ valuable comment on the title of the study. As suggested, we have modified the study title as follows:
Efficacy and Safety of Sesame Oil Cake Extract on Memory Function Improvement:
A 12-week, randomized, double-blind, placebo-controlled pilot study.
Also, we have modified the study sample size as follows:
The number of subjects needed in each group to achieve 80% power for a 5% significance level with a two-sided test was 25 persons per group. A total of 70 subjects (35 subjects per group) was required assuming a dropout ratio of 30%. Therefore 70 subjects were enrolled for 1:1 randomization to the SOCE and placebo groups.
Thank you!
